# TNFR1 Suppression by XPro1595 Reduces Peripheral Neuropathies Associated with Perineural Invasion in Female Mice

**DOI:** 10.3390/cells14221749

**Published:** 2025-11-07

**Authors:** Morgan Zhang, Naijiang Liu, Kesava Asam, Charles Meng, Bradley Aouizerat, Yi Ye

**Affiliations:** 1Translational Research Center, New York University College of Dentistry, New York, NY 10010, USA; 8372651@gmail.com (M.Z.); nl2782@nyu.edu (N.L.); kra277@nyu.edu (K.A.); bea4@nyu.edu (B.A.); 2Pain Research Center, Department of Molecular Pathobiology, New York University College of Dentistry, New York, NY 10010, USA; 3Doctor of Dental Surgery Degree Program, New York University College of Dentistry, New York, NY 10010, USA; qm316@nyu.edu; 4Department of Biomedical Engineering, Tandon School of Engineering, New York University, New York, NY 10010, USA

**Keywords:** perineural invasion, pain, tumor necrosis factor, XPro1595, mitochondrial, Schwann cell, myelination, extracellular matrix, inflammation, locomotion

## Abstract

**Highlights:**

**What are the main findings?**
In a mouse model of perineural invasion, blocking soluble TNFα signaling with XPro1595—but not TNFR1 knockout—reduced tumor burden, mechanical allodynia, and locomotor deficits, primarily in females.XPro1595 may exert its beneficial effects via promoting mitochondrial function and myelination while suppressing inflammatory, extracellular matrix, and tumor progression pathways.

**What is the implication of the main finding?**
Pharmacologically targeting soluble TNFα with XPro1595 may represent a promising, sex-dependent therapeutic strategy to alleviate cancer-associated pain and nerve injury linked to PNI.TNFα signaling contributes to both tumor progression and nerve pathology in a sex-specific manner, highlighting the need for precision approaches in cancer pain management.

**Abstract:**

Perineural invasion (PNI), defined by cancer spreading or invading into the nerve, links to severe pain, recurrence, and poor prognosis. PNI contributes to nerve damage, Schwann cell activation, and sensory neuron dysfunction. Soluble tumor necrosis factor α (solTNFα) binds to TNFR1 to drive inflammation and nerve injury, playing a key role in cancer progression and pain. This study, using a mouse sciatic nerve PNI model, explored whether blocking solTNFα-TNFR1 signaling via TNFR1 knockout or pharmacological inhibition by XPro1595 could reduce PNI-associated pain. Data showed that XPro1595, but not TNFR1 knockout, reduced tumor burden, alleviated mechanical allodynia, and improved muscle function and locomotion, primarily in females. Histological analysis in females showed that XPro1595 increased the number of myelin and dendritic cells while reducing axonal damage that resulted from PNI. In the tumor zone outside the nerve truck, XPro1595 reduced T cell and increased macrophage and dendritic cell numbers. Transcriptomic analysis revealed that XPro1595 in females with PNI upregulated mitochondrial, myelination, motor function, and immune regulation gene pathways while it downregulated inflammatory, extracellular matrix, and tumor progression pathways. Overall, we demonstrated that XPro1595 exhibited antitumor, neuroprotective, and analgesic properties in female mice, likely by promoting neuronal regeneration and mitochondrial function, while reducing inflammation and extracellular remodeling.

## 1. Introduction

Perineural invasion (PNI), defined as the presence of tumors in close proximity to a nerve or the presence of tumor cells within the nerve, is linked to increased pain, locoregional recurrence, and reduced survival rates [1,2]. PNI predominantly occurs in head and neck cancer (HNC). Our recent research demonstrated that PNI is an independent predictor of function-evoked pain, even after controlling for demographic and other clinicopathological variables in a multi-center study in patients with HNC [3]. In mouse models of HNC PNI, PNI-induced nociceptive behaviors and motor dysfunction, which recapitulates the pain reported by patients with PNI [4]. The reported mechanisms include extracellular matrix (ECM) remodeling, inflammation, nerve damage, mitochondrial dysfunction, and altered sensitivity of primary afferent neurons [5,6].

TNFα is a cytokine that has pleiotropic effects on various cell types. It has been identified as a major regulator of inflammatory responses [7], ECM remolding [8], nerve demyelination [9], mitochondrial oxidative stress [10,11], and neuronal sensitization [12,13,14,15]. It has been implicated as a mediator of cancer progression [16,17] and pain in HNC [18,19,20,21]. Tumor necrosis factor α (TNFα) exists in both a soluble form (solTNFα) and a transmembrane form (tmTNFα) [22]. solTNFα preferentially binds to tumor necrosis factor receptor 1 (TNFR1), while tmTNFα preferentially binds to TNFR2 [22]. It has been proposed that the solTNFα–TNFR1 interaction is pro-inflammatory and nociceptive, whereas the tmTNFα–TNFR2 interaction is neuroprotective and anti-inflammatory [23,24]. Consistent with this hypothesis, elevated concentrations of solTNFα are frequently detected in tumor tissues, blood, and saliva samples from cancer patients [25] and may promote cancer cell invasiveness and metastasis through autocrine and paracrine signaling [16,26]. The solTNFα–TNFR1 interaction drives and sustains neuropathic pain in animal models of nerve injury [24,27,28]. Tumor-derived solTNFα correlates with oral cancer pain in patients [19]. In animal models of tongue cancer, solTNFα was shown to amplify oral cancer pain through its effects on cancer cells, sensory neurons, Schwann cells (SCs), immune cells, and sympathetic modulation [20,21]. TNFR1 knockout (KO) has been shown to affect the behavioral responses in non-tumor mice [29,30,31] and reduce neuropathic pain [28,32].

In the present study, we examined whether blocking solTNFα-TNFR1 signaling by TNFR1 gene deletion or selective pharmacological inhibition using XPro1595 can reduce PNI or PNI-associated pain. In wild-type mice and mice with global TNFR1 KO, we produced a syngeneic model of PNI by inoculating mouse oral cancer cells into the sciatic nerve [33,34]. Currently, all approved anti-TNFα drugs are non-selective and block both TNFR1 and TNFR2 signaling [35]. To selectively sequesters solTNFα without affecting tmTNFα, we used XPro1595, a clinical-grade dominant-negative TNFα biologic developed by INmmunoBio that sequesters solTNF into inactive heterotrimers, thereby reducing TNFR1-biased pro-inflammatory signaling while sparing tmTNF/TNFR2 [36]. XPro1595 is currently in Phase I clinical trials for Alzheimer’s disease [37]. In our study, we found that XPro1595 reduced mechanical pain and tumor progression as well as improved motor function primarily in female mice. XPro1595 reversed the tumor-induced decrease in axons and SCs, and suppressed Tregs, and CD4^+^ and CD8^+^ T cells. XPro1595 downregulated distinct functional clusters of differentially expressed genes (DEGs) that supported its antinociceptive effect. In contrast, TNFR1 KO did not significantly alter tumor-induced pain, motor function change, and its growth. Despite that, TNFR1 KO affected the behavioral responses in non-tumor mice at the baseline.

## 2. Materials and Methods

### 2.1. Mice

Male and female wide-type (WT) C57BL/6J (#000664) and global TNFR1 KO mice (C57BL/6-TNFrsf1atm1Imx/J, #003242) were obtained from The Jackson Laboratory. The studies involving these mice were approved by the NYU Institutional Animal Care and Use Committee. Mice (aged 2 to 3 months) were randomly assigned to different groups. Behavioral assays were conducted with blinding. Both TNFR1 KO mice and XPro1595 treatment were used to control for developmental confounds inherent to gene knockout models and to avoid off-target effects of pharmacological inhibition.

### 2.2. The Mouse PNI Model

The PNI models were generated as we previously reported [33,38]. Under anesthesia, three µL of media (the sham group) or media containing 5000 mouse oral cancer (MOC2, kerafast, EWL002—FP) cells were injected into the right sciatic nerve. Four groups were used: sham (*n* = 14/sex), vehicle (saline) treated WT tumor-bearing control mice (Con, *n* = 14), KO (TNFR1^−/−^ tumor-bearing mice, *n* = 10 males and 14 females), and XPro (XPro1595-treated WT tumor-bearing mice, *n* = 12 males and 10 females). XPro1595 (donated by INmune Bio Inc., Boca Raton, FL, USA, 10 mg/kg) or saline was injected subcutaneously [39], at post-inoculation day (PID) 8, 10, 13, and 15. Sample size was determined based on pilot studies, assuming an effect size of 1.4, power = 0.8, a = 0.05. A minimum of *n* = 10/group is needed.

### 2.3. Von Frey Paw Withdrawal Assay

Mice were individually placed in transparent boxes with mesh floors and acclimated for one hour before measurement. Mechanical nociception was assessed using von Frey filaments (Stoelting, Wood Dale, IL, USA) [40], measured at the baseline, PID 8 and PID 14.

### 2.4. Hargreaves’s Test

Paw withdrawal latency was measured using the Hargreaves’ Apparatus [41]. An average of 3–4 trials per animal were taken >5 min apart at the baseline, PID 8 and PID 14. A cutoff was set at 20 s for heat stimulation to avoid tissue damage.

### 2.5. Behavioral Spectrometer

The behavioral spectrometer (Viewer3, BiObserve, Bonn, Germany) [42,43], is a non-operative device that automatically records voluntary behaviors including running, walking, trotting, track length, ambulation, and time spent in the inner zone. Mice were recorded for 15 min at the baseline as well at PID 15.

### 2.6. Toe-Spread Assessment

Toe-spread reflex has been used to identify potential motor deficits including muscle weakness and impaired motor control [33,44]. Mice were lifted by the tail to observe toe spread in their hind paws. Digit abductions were scored on a five-point scale (0 = normal; 4 = maximal reduction in digit abduction and leg extension).

### 2.7. Tumor Size Measurement

On PID 17, following the toe-spreading test, the mice were sacrificed and sciatic nerves were collected. Tumor size was measured with a digital caliper, calculated using the formula AB^2^/2, where A and B represent the maximum length and width, respectively.

### 2.8. Multiplex Immunohistochemistry and Immunofluorescence (mIHC/IF) Analysis

After fixation in 10% formalin at 4 °C for 48 h, sciatic nerve tissues were dehydrated through graded ethanol solutions and xylene, followed by infiltration with paraffin (Paraplast X-tra, Leica, Wetzlar, Germany, SKU 39603002) using a Leica Peloris II tissue processor. The embedded tissues were sectioned transversely at a thickness of 5 µm, starting from the midpoint of the tumor if an enlarged tumor was present. The sections were then stained with hematoxylin (Leica, Wetzlar, Germany, 3801575) and eosin (Leica, Wetzlar, Germany, 3801619) using a Leica ST5020 automated stainer to assess histological features. Tumor area was further measured by the cross-section of the nerve based on hematoxylin/eosin staining, which is similar to a measurement performed in the clinic. Adjacent sections were stained using Akoya Biosciences^®^ Opal™ (Marlborough, MA, USA) multiplex immunofluorescence reagents on a Leica BondRx autostainer, following the manufacturer’s instructions. The sections were initially treated with peroxide to inhibit endogenous peroxidases, followed by antigen retrieval using ER2 (Leica, Wetzlar, Germany, 9AR9640) for 20 min at 99 °C. The slides were then incubated with the first primary antibody and a secondary polymer pair (Rabbit-on-Rodent HRP polymer, Cat: RMR622, Biocare, Pacheco, CA, USA), followed by HRP-mediated tyramide signal amplification with a specific Opal^®^ fluorophore. The primary and secondary antibodies were subsequently removed through a heat retrieval step, leaving the Opal fluorophore covalently linked to the antigen. This sequence was repeated with subsequent primary and secondary antibody pairs, each utilizing a different Opal fluorophore. Due to the constraints of the current Leica 7-Color Double Dispensing protocol, each slide was limited to staining with no more than six primary antibodies. Sections were counterstained with spectral DAPI (Akoya Biosciences, Marlborough, MA, USA, FP1490) and mounted using ProLong Gold Antifade (ThermoFisher Scientific, Waltham, MA, USA, P36935). Semi-automated image acquisition was conducted on an Akoya Vectra Polaris (PhenoImagerHT, Akoya Biosciences, Marlborough, MA, USA) multispectral imaging system at 20× or 40× magnification, utilizing PhenoImagerHT 2.0 software in conjunction with Phenochart 2.0 and InForm 3.0 to generate spectrally unmixed whole-slide QPTIFF scans.

The antigens for the lymphoid cell markers included Ly6G (clone #: 1A8 (RUO), Catalog # 551459, Vendor: BD, dilution 1:400), NF200 (NB300-135, Novus Bio, dilution 1:800), FOXP3 (D6O8R, #12653, CST, dilution 1:4000), CD4 (D7D2Z, 25229, CST, dilution 1:300), and CD8 (D4W2Z, #98941, CST, dilution 1:300). For the myeloid cell markers, the antigens included F4/80 (D2S9R, #70076, CST, dilution 1:1000), Ly6C (EPR27220-23, ab314120, Abcam, dilution 1:1000), CD4 (D7D2Z, #25229, CST, dilution 1:300), CD8 (D4W2Z, #98941, CST, dilution 1:300), CD68 (E3O7V, #97778, CST, dilution 1:300), and CD11b (NB110-89474, Novus, dilution 1:4000). Lastly, the antigens for the nervous cell marker panel (P3) included MPZ (EPR20383, ab183868, Abcam, dilution 1:500), FOXP3, and GFAP (ab7260, Abcam, dilution 1:4000).

A quantitative analysis of images obtained through mIF staining for each area was conducted as previously described, with minor modifications [45,46]. Cell segmentation, thresholds, and regions of interest (ROIs) were configured using QuPath Analysis Software (version 0.4.4) [47]. Cell counts and the percentage of the ROI exhibiting signal were measured after thresholding the grayscale image for each channel. Statistically optimal thresholds for staining intensity were generally calculated using the triangle algorithm [48]. The same thresholding parameters were applied consistently to all samples on the same slide.

### 2.9. RNA Sequencing

Total RNA was extracted from fresh-frozen control mouse sciatic nerve tissues, both with and without tumors, using the Qiagen RNeasy kit (Catalog No: 74104). The RNA concentration and integrity were assessed using an Agilent 2100 Bioanalyzer with an RNA 6000 Nano kit (Part No: 5067-1511). Ribo-depleted, strand-specific RNA libraries were prepared from PNI samples (*n* = 3/group). Sequencing reads were generated in-house and uploaded to the NYU Greene HPC. Quality control and adapter trimming were performed with fastp (v0.20.1). Cleaned reads were aligned to the mouse reference genome GRCm39 using HISAT2 (v2.2.1). Alignments were converted to sorted BAM files with SAMtools (v1.12). Transcript assembly and gene-level quantification were performed with StringTie (v2.1.6) to produce a gene count matrix.

Downstream analyses were conducted in R (v4.3.0). Genes were retained if they had at least 10 counts in at least 90% of samples. Ensembl gene identifiers were mapped to Entrez Gene IDs using EnrichmentBrowser (v2.30.1). Differential expression was performed with DESeq2 (v1.40.1). Surrogate variables were estimated with sva (v3.48.0) and included as covariates in the design. Log2 fold-change shrinkage was applied with lfcShrink using the apeglm method (apeglm v1.22.1) to obtain stabilized effect-size estimates. DEGs used for pathway analysis met the following thresholds: adjusted *p* value < 0.05 and log2 fold change < −0.75 or >0.75. A total of 905 upregulated and 475 downregulated genes satisfied these criteria.

The VolcaNoseR web server was utilized to create and label volcano plots. The *x*-axis represents log_2_ fold change, while the *y*-axis represents −log_10_ adjusted *p*-value. “Top hits” were ranked based on Manhattan distance (|ΔX|  + |ΔY|) from the origin [49]. Pathway analysis was conducted using the DAVID 2021 tool [49] in conjunction with the Expression Analysis Systematic Explorer (EASE) to examine biological processes (BPs), cellular components (CCs), molecular functions (MFs), and Kyoto Encyclopedia of Genes and Genomes (KEGG) pathways. Upregulated and downregulated DEGs were analyzed separately, as analyzing all DEGs together may obscure significant directional changes [50]. Functional annotation parameters for upregulated DEGs are as follows: similarity term overlap = 4; initial group membership = 5; similarity threshold = 0.5; final group membership = 5; multiple linkage threshold = 0.5; enrichment threshold (EASE) = 0.05. Functional annotation parameters for downregulated DEGs are as follows: similarity term overlap = 3; initial group membership = 4; similarity threshold = 0.5; final group membership = 3; multiple linkage threshold = 0.5; enrichment threshold (EASE) = 0.05.

### 2.10. Statistical Analysis

Statistical analyses were conducted using Prism 9.0 (GraphPad Software). For comparisons between two groups, Student’s *t*-test was employed. Multiple group comparisons were performed using one-way ANOVA with Dunnett’s post hoc analysis or two-way ANOVA followed by Holm–Sidak post hoc tests. Statistical significance was set at *p* < 0.05. Results were expressed as mean ± standard deviation (SD).

## 3. Results

### 3.1. TNFR1 Gene Deletion Affects Sensory and Motor Function in a Sex Dependent Manner in Tumor-Free Mice

Since TNFR1 global gene knockout could affect baseline mouse behaviors [29,30,31], we conducted von Frey tests for mechanical allodynia and Hargreaves’s test for thermal hyperalgesia in both male and female WT and KO mice. Additionally, the behavioral spectrometer box was used to assess anxiety-like behaviors (time spent in the inner zone) and locomotion activities (e.g., running, walking, and trotting).

For sensory behaviors, TNFR1 knockdown significantly increased both mechanical withdrawal thresholds and thermal paw withdrawal latency compared to the WT (Figure 1A,B) in female, but not male, mice. In the behavioral spectrometer, TNFR1 knockdown did not affect time spent in the inner zone compared to the WT (Figure 1C) in both females and males. For the locomotor behaviors (Figure 1D,F), TNFR1 knockdown significantly reduced total track length and tended to reduce both walk time and trot time compared to the WT in females, whereas TNFR1 knockdown did not show any effects in males. This data demonstrates that TNFR1 gene deletion influences sensory and motor behaviors in a sex-dependent manner at baseline, with notable effects observed predominantly in female mice. 

### 3.2. TNFR1 Gene Deletion or XPro1595 Treatment Affects Sensory and Motor Function in a Sex Dependent Manner in Mice with PNI

To determine the role of TNFR1 signaling in PNI-associated sensory and motor defects, we inoculated MOC-2 cells into the sciatic nerve of WT and TNFR1 KO mice. In WT mice with PNI, repeated doses (10 mg/kg [28,51], see Figure 2A) of XPro1595 were administered to block TNFR1 signaling. For behaviors that showed a baseline difference in KO mice, a percentage change from baseline was used for group comparisons. In female WT mice, PNI significantly reduced mechanical withdrawal thresholds observed on post-inoculation day (PID) 8 and 14 (Figure 2C) and thermal paw withdrawal latency at PID 14 compared to the sham group (Figure 2E). Female TNFR1 KO mice with PNI exhibited less mechanical pain at PID 8 but not at PID 14. In contrast, XPro1595 treatment in females reduced PNI-induced mechanical pain at both PID 8 and PID 14 (Figure 2C). However, neither TNFR1 KO nor XPro1595 affected the PNI-induced thermal pain in females. These results indicate that, in female mice, PNI induced mechanical allodynia and thermal hyperalgesia, while TNFR1 blockade, and to a lesser degree, gene KO alleviated, mechanical allodynia but not thermal hyperalgesia. In male mice, PNI did not significantly alter mechanical withdrawal thresholds (Figure 2B), but reduced thermal paw withdrawal latency on PID 14 compared to the sham group (Figure 2D). TNFR1 KO or XPro1595 had no significant effect on mechanical or thermal measures in males. No significant difference in time spent in the inner zone was observed across different treatment groups or sex (Figure 2F).

Regarding motor functions, we found in females that PNI significantly reduced walk time, trot time, run time, track length, ambulation, and toe-spreading scores (Figure 2G–L) compared to the sham group. XPro1595 significantly improved all these PNI-induced motor defects, whereas TNFR1 KO had no effect. In males, PNI showed less effects on motor functions, only reduced run time and ambulation (Figure 2I or Figure 2K). XPro1595 improved PNI-induced reduction in ambulation (Figure 2K) but not run time (Figure 2I), whereas TNFR1 KO had no effect on any of these motor function-related outcomes. These results indicated that PNI-induced impairment in motor functions was more severe in females. XPro1595 reversal of PNI-induced motor defects was also primarily observed in females.

### 3.3. Xpro1595 Reduced Nerve Damage, Neuroinflammation, and Tumor Size

Since XPro1595 effectively ameliorated PNI-associated symptoms in female mice, we performed the multiplexed immunofluorescence staining in female mice to understand whether XPro1595 attenuates PNI-associated nerve injury and neuroinflammation. We found that compared to sham-operated mice, PNI significantly decreased NF200^+^ areas (axons, Figure 3D,E), MPZ^+^ areas (myelin, Figure 3H,I), GFAP^+^/MPZ^+^ areas (likely repair or activated Schwann cells, Figure 3H or Figure 3J). PNI marginally reduced GPAP^+^ only areas (nonmyelinated SCs, Figure 3H or Figure 3M) while significantly increased the cell number of CD11b^+^/DAPI^+^/F4/80^−^/Ly6C^−^ (likely a subset of monocytes or immature myeloid cells, Figure 3K,L) and CD68^+^/NF200^+^ area (Figure 3F,G). XPro1595 treatment, however, significantly increased the NF200^+^ area (axons, Figure 3D,E), MPZ^+^ area (myelinated SCs, Figure 3H,I), GFAP^+^/MPZ^+^ area (Figure 3H or Figure 3J), CD68^+^/NF200^+^ area (Figure 3F,G), as well as the cell number of CD11b^+^/F4/80^−^/Ly6C^−^ immune cells (Figure 3K,L). In the nerve area, we did not detect any F4/80^+^/CD11b^+^ macrophages, Ly6G^+^ neutrophils, or Ly6C^+^/DAPI^+^/CD11b^−^/F4/80^−^ cells among the three mouse groups. The tumor area was smaller in the XPro1595 treatment group (Figure 2M).

### 3.4. XPro1595 Treatment Altered Tumor Immune Environment

In the tumor zone outside the nerve identified by the tumor cell’s distinctive large blue nuclei, shown in H&E staining, as well as the lacking of NF200 or MPZ nerve markers, we found that the tumor microenvironment exhibited an increased number for FOXP3^+^/CD4^+^ Tregs (Figure 4A,B), CD4^+^ T cells (Figure 4A or Figure 4C), CD8^+^ T cells (Figure 4D,E), Ly6G^+^ neutrophils (Figure 4H,I), CD11b^+^/F4/80^−^/Ly6C^−^ (Figure 4J,K), and Ly6C+/CD11b^−^/F4/80^−^ (Figure 4J or Figure 4L) cells. XPro1595 significantly decreased the number of Tregs (Figure 4A,B), CD4^+^ T cells (Figure 4A or Figure 4C), and CD8^+^ T cells (Figure 4D,E), but significantly increased the number of F4/80^+^/CD11b^+^ macrophages (Figure 4F,G), and CD11b^+^/F4/80^−^/Ly6C^−^ (Figure 4J,K) cells.

### 3.5. Transcriptomic Insights on the Analgesic Mechanism of XPro1595

To elucidate further the molecular mechanisms of beneficial effects exerted by XPro1595 in female mice with PNI, we performed RNA-seq in sciatic nerve tumor samples from female mice treated with and without XPro1595. In total, we identified 1390 DEGs including 914 upregulated and 476 downregulated genes (Appendix A).

#### 3.5.1. Upregulated DEGs and Functional Clusters

At the individual gene level, the top 20 hits among upregulated DEGs (Figure 5A), detected and ranked based on Manhattan distance by DAVID, included *Serping1*, *Hp*, *Apod* (Cluster 1), *Mmp3* (Cluster 2), *C4b* (Cluster 1), *C3* (Cluster 5), *Pmp22*, *Cilp*, *Mpz*, *Gapdh* (Cluster 5), *Serpina3n*, *Rcn3*, *Lox*, *Trim63*, *Mfap4*, *C1qTNFα3*, *Cox7b* (Cluster 1), *Fabp4*, and *Igfn1*. Gene clusters are presented in Figure 5B. *Cox7b* encodes for a component of cytochrome c oxidase, the final enzyme in the mitochondrial electron transport chain that drives oxidative phosphorylation. *Fabp4* encodes for a fatty acid-binding protein involved in fatty acid uptake, transport, and metabolism. *Fabp4* is also part of the PPAR signaling pathway [52,53] and may mitigate mitochondrial dysfunction by reversing excessive fatty acid oxidation or correcting mitochondrial defects in neurodegenerative diseases [54]. *Apod* encodes for a component of high-density lipoprotein and functions as a transporter of small hydrophobic molecules, primarily associated with lipid metabolism and neuroprotection [55]. Both *Pmp22* and *Mpz* encode for myelin proteins produced by Schwann cells that are negatively regulated by TNFα and mechanical allodynia. Loss of function or decreased expression of these myelin proteins contribute to demyelination diseases and neuropathic pain [56,57,58].

For the upregulated DEGs, the top six enriched clusters (Figure 5B) include: cytosolic ribosome (Cluster 1), structural constituent of chromatin (Cluster 2), chemokine activity (Cluster 3), mitochondrial dysfunction in neurodegenerative diseases (Cluster 4), myosin II complex (Cluster 5), and positive regulation of neutrophil chemotaxis (Cluster 6). Particularly, Clusters 3–6 supported our hypothesis and are consistent with findings from our behavioral and immunohistological studies.

Cluster 3 is composed of 42 genes, including several chemokine ligands as pivotal hub genes, such as *Ccl12*, *Ccl2*, *Ccl7*, *Ccl11*, *Ccl17*, *Ccl21d*, *Ccl21b*, *Ccl8*, *Ccl21a*, and *Ccl21b*. This cluster is involved in the antimicrobial humoral immune response mediated by antimicrobial peptides (Biological pathway, BP category), chemokine activity (Molecular function, MF category), and viral protein interaction with cytokines and cytokine receptors (KEGG category).

The mitochondrial dysfunction in the neurodegenerative diseases cluster (Cluster 4) comprises 95 genes, including several key mitochondrial protein genes acting as pivotal hub genes, such as *Sdhb*, *Ndufa4*, *Ndufb4c*, *Ndufb4b*, *Ndufv3*, *Ndufa9*, *Ndufc1*, *Ndufs5*, *Ndufb4*, *Ndufa12*, *Ndufa11*, and *Ndufb1*. This cluster is involved in proton motive force-driven mitochondrial ATP synthesis (BP category), mitochondrial inner membrane function (cellular components, CC category), and pathways related to prion disease and Huntington’s disease (KEGG category). This cluster also includes upregulated *Pparg*, a transcription factor that suppresses TNF expression and is neuroprotective, anti-inflammatory, and anti-mitochondrial dysfunction [40,52]; the dysfunction of which is implicated in multiple neurodegenerative diseases [59,60,61].

The upregulation of genes in Cluster 5 supports the observed recovery of motor function following XPro1595 administration. Cluster 5 consists of 22 genes, with several myosin subunits serving as pivotal hub genes, including *Myo18b*, *Myh1*, *Myh11*, *Myh2*, *Myh4*, and *Myh8*. This cluster is primarily associated with myofibril structure (CC category) and cytoskeletal motor activity (MF category).

Cluster 6 (12 genes) is primarily involved in the positive regulation of neutrophil chemotaxis (BP category). This cluster includes chemokine ligand and receptor genes (*Ccl21a*, *Ccl21b*, *Ccl21d*, *Ccr7*), *C3ar1* (complement component 3a receptor 1), *Il1b* (interleukin 1 beta), *Sell* (selectin, lymphocyte), *Lbp* (lipopolysaccharide binding protein, which facilitates the interaction of LPS with CD14 and TLR4, triggering an immune response), and *Thbs4* (thrombospondin 4, which influences inflammatory responses and the expression of certain chemokines).

#### 3.5.2. Downregulated DEGs and Their Functional Clusters

The top six enriched clusters (Figure 5C) of downregulated DEGs were cardiac muscle cell action potential (Cluster 1), cell–cell junction assembly (Cluster 2), excessive inflammation and protein serine kinase activity (Cluster 3), presynaptic active zone (Cluster 4), basement membrane (Cluster 5), and calcium ion transport (Cluster 6). The top 20 hits in downregulated DEGs (Figure 5A), detected and ranked by Manhattan distance in DAVID, are *Snord15b*, *Nav2*, *Neat1*, *Chka* (Cluster 3), 9230105E05Rik, *Ltbp3* (Cluster 5), *Firre*, *Clcf1*, *Irs1*, *Zfhx3*, *Elf3*, *Muc4* (Cluster 5), *Pitpnm3*, *Gm24265*, *Mgat3*, *2610035D17Rik*, *Als2cl*, *Tnnt2*, and *Kcnq1ot1*. The downregulation of genes in Cluster 3 supports the anti-inflammatory effect of XPro1595 treatment (Figure 3). Cluster 3 is composed of 36 genes, including 9 protein kinase genes: *Map3k9*, *Camk1g*, *Prkcz*, *Wnk4*, *Lmtk3*, *Mapk13*, *Ksr2*, *Hunk*, and *Prkcg* serving as hub genes. Excessive protein activity of serine kinases, particularly mitogen-activated protein kinases (e.g., *Map3k9* and *Mapk13*) and protein kinase C (e.g., *Prkcz* and *Prkcg*) can lead to excessive inflammation by triggering the overproduction of pro-inflammatory cytokines and mediators [62,63]. Cluster 3 is primarily involved in protein serine kinase activity and calcium channel activity in the MF category.

The anti-inflammatory effect of XPro1595 may also be linked to the downregulation of a “top hit” gene, *Elf3*. *Elf3* encodes for a DNA-binding transcription activator. Inflammatory stimuli, including TNFα, can induce its expression in various cell types, subsequently activating genes involved in chemokine and cytokine production, as well as inflammation-promoting matrix metalloproteinases [64].

The downregulation of genes in Clusters 4 and 6 supports our finding of the antinociceptive effect of XPro1595 treatment (Figure 1A,B). Cluster 4 is composed of nine genes, with three presynaptic cytoskeletal matrix protein genes, *Pclo*, *Unc13a*, and *Unc13,* as pivotal hub genes. *Unc13a* is essential for synaptic vesicle maturation in most excitatory/glutamatergic, but not inhibitory/GABA-mediated, synapses. Cluster 4 is primarily involved in the presynaptic active zone cytoplasmic component in the CC category. Cluster 6 is composed of eight genes, with four TRP channel genes (*Trpv3*, *Trpv4*, *Trpv6*, and *Trpm4*) and subunit 1 of the NMDA receptor gene (*Grin1*) as pivotal hub genes. Notably, NMDA and TRPV1 receptors interact in trigeminal sensory neurons to mediate mechanical hyperalgesia [65]. Cluster 6 is primarily involved in ion channels that are important in regulating neuronal excitability and sensitization [66,67].

The antitumor effect of XPro1595 treatment was observed in tumor size measurements (Figure 2M) and immunohistological studies (Figure 3A,B). This effect may be attributed to the downregulation of several “top hit” genes, including *Snord15b* [68], *Nav2* [69], *Neat1* [70], *Firre* [71], *Tnnt2* [72], and *Kcnq1ot1* [73], which may function as oncogenes that promote the progression of several solid tumors. This effect may also be linked to the downregulation of the “top hit” genes *Chka* and *Muc4*. *Chka* encodes for a protein that plays a key role in phospholipid biosynthesis and may contribute to tumor cell growth [74]. *Muc4* encodes a highly glycosylated protein that constitutes a major component of mucus and may promote tumor growth by repressing apoptosis [75].

The downregulation of genes in Cluster 5 supports our finding of the ECM remodeling effect in our previous study that PNI is associated with function-evoked pain and altered ECM [3]. Cluster 5 comprises 13 genes, including 6 collagen-containing extracellular matrix protein genes—*laminin*, *alpha 5* (*Lama5*), *agrin* (*Agrn*), *collagen*, *type VII*, *alpha 1* (*Col7a1*), *collagen*, *type IX*, *alpha 3* (*Col9a3*), *collagen*, *type XVII*, *alpha 1* (*Col17a1*), *collagen*, *type IV*, *alpha 4* (*Col4a4*)—which serve as pivotal hub genes. Cluster 5 is involved in collagen-containing ECM and basement membrane in the CC category, and ECM structural constituent conferring tensile strength in the MF category.

## 4. Discussion

In the present study, we demonstrated that pharmacological inhibition of soluble TNF/TNFR1 signaling by XPro1595, but not TNFR1 gene deletion, significantly reduced PNI-associated mechanical hypersensitivity, mitigated tumor progression, and improved motor performance primarily in female mice. Mechanistically, XPro1595 reduced tumor-induced nerve damage, altered the neural–immune tumor microenvironment, and regulated broad transcriptional networks, including gene clusters associated with mitochondrial function, immune regulation, myelination, ECM remodeling, neuronal, and tumor growth.

TNF is a central mediator of pathological pain, acting predominantly through TNFR1. In the periphery, TNFR1 signaling sensitizes nociceptors by upregulating excitatory ion channels such as TRPV1 and Nav1.8, thereby increasing neuronal excitability and promoting release of pro-nociceptive neuropeptides, including substance *p* and calcitonin gene-related peptide (CGRP) [76,77]. Within the central nervous system, TNFR1 activation on microglia and astrocytes initiates inflammatory cascades mediated by p38 MAPK, NF-κB, and JNK. This leads to the release of additional cytokines (e.g., TNFα, IL-1β, IL-6), phosphorylation of NMDA receptors, enhancement of excitatory synaptic transmission, and suppression of inhibitory GABAergic tone [78,79]. Beyond sensitization, TNFR1 contributes to neuronal apoptosis and maladaptive plasticity, thereby sustaining neuropathic pain states [80].

Despite its recognized pro-nociceptive role, genetic studies demonstrate that TNFR1 signaling may exert context-dependent effects, influenced by disease model and sex. For example, our data showed that TNFR1 KO increased mechanical thresholds and thermal latency and reduced locomotor activity in tumor-free female, but not male, mice. Previous studies reported that TNFR1 KO does not alter baseline mechanical or thermal sensitivity [81,82], but these analyses lacked sex-specific comparisons. In our PNI model, TNFR1 KO did not mitigate pain or motor dysfunction. These findings align with previous reports that TNFR1 KO does not alter PNI-induced changes in afferent excitability [5]. The ineffectiveness of TNFR1 deletion in PNI pain may stem from developmental compensation, whereby alternative signaling pathways replace TNFR1 function during development. It has been shown that TNFR1 is important for nociceptor development, regulating TrkA^+^ peptidergic nociceptors and Ret^+^ non-peptidergic nociceptors differently [83]. TNF/TNFR1 can be antinociceptive [83]. In cancer, TNF/TNFR1 has been shown to mediate both pro-survival and pro-apoptotic signaling pathways [26,84,85,86,87,88]. Genetic deletion during development may disrupt the complex TNFR1 signaling networks, thereby neutralizing potential analgesic benefits.

Unlike TNFR1 KO, XPro1595 robustly attenuated PNI-associated pain and improved motor outcomes in females. This is consistent with findings in multiple pain models. For example, in a mouse acute pancreatitis model, XPro1595 blunted disease severity and the associated referred mechanical hypersensitivity. Mechanistically, XPro1595 prevented pancreatic inflammation and acinar cell death, and reduced hippocampal astrocyte reactivity induced by pancreatitis [39]. In an orofacial inflammatory pain model, peripheral delivery of an XPro1595 prevented CFA-evoked hyperalgesia by suppressing TRPV1 upregulation and nociceptor sensitization [89]. XPro1595 ameliorated bone cancer pain via inhibiting glial cell activation and neuroinflammation [51]. In a spared nerve injury (CCI) neuropathic pain model, XPro1595 accelerated recovery from mechanical hypersensitivity in male mice only. The treatment reduced elevated NMDA receptor levels in the brains of males. In females, CCI reduced NMDA receptors and estrogen impairs XPro1595 analgesia efficacy by modulating NMDA receptors [28]. This discrepancy may be explained by model-specific regulation of NMDA receptors: CCI reduces NMDA receptor levels [28], whereas oral cancer upregulates NMDA receptors in females (unpublished data, under revision).

The improvement in pain and motor outcomes by XPro1595 in mice with PNI can be attributed to its neuroprotective, immunomodulatory, and antitumor effects. PNI is associated with altered ECM, nerve damage, mitochondrial dysfunction, inflammation, sensory and motor dysfunction [3,4,32]. Transcriptomic profiling revealed that XPro1595 treatment upregulated pathways supporting mitochondrial function, myelination, immune regulation, and motor function, whereas downregulated pathways in inflammatory cascades, ECM remodeling, and tumor progression. At the cellular level, XPro1595 modulated the immune landscape in PNI. While PNI increased regulatory T cells, pro-inflammatory CD4^+^ and CD8^+^ effector T cells, neutrophils, and CD11b^+^ myeloid cells, XPro1595 treatment selectively reduced regulatory T cells and effector T cells, while increasing macrophages and CD11b^+^ cells. These shifts in immune subsets may collectively contribute to its analgesic and antitumor properties despite opposing effects exerted by immune suppressive and cytotoxic immune cells [90]. XPro1595 has been shown in other models to reduce age-dependent accumulation of activated immune cells and CD4^+^ T cell [91].

Using the PNI model, we identified a robust sex difference in pain behavior. PNI-induced mechanical hypersensitivity was observed exclusively in females, while thermal hypersensitivity and motor impairments were observed in both sexes, with worse outcomes in females. These results parallel clinical findings in cancer patients, where PNI is associated with increased pain burden and independently predicts function-evoked pain [3,33]. Although clinical studies have not consistently shown sex differences in PNI-related pain, it is well established that chronic pain, including oral cancer pain, is more prevalent and severe in women [92,93]. Importantly, pain is a multifaceted experience characterized by sensory modality (mechanical, thermal, and chemical), quality (burning, tingling, dull, and sharp), temporal dynamics, and frequency. HNC patients often experience burning pain and increased sensitivities to sour and spicy food, in addition to mechanical allodynia [94]. However, most preclinical and clinical studies, including ours, evaluate only a subset of these dimensions. This limitation complicates interpretation of sex differences across models.

Sex differences in pain perception are well established, with females generally showing greater sensitivity than males. Estrogen plays a central role, as it can both dampen the antinociceptive effects of TNFR1 inhibition [28,32] and enhance TNFα/TNFR1 signaling [95], promoting neuronal sensitization to inflammation [96,97,98]. These findings suggest that TNFR1-mediated pathways may contribute more strongly to pain in females, including in perineural invasion, where estrogen upregulates TNFα/TNFR1 signaling in some tumor cells. Although our study did not directly assess estrogen’s influence, this remains an important direction for future research on hormonal regulation of neuroinflammatory pain.

## 5. Conclusions

Our study demonstrates that XPro1595 is a robust immune modulator and may exert analgesic, antitumor, and neuroprotective effects in a sex and context dependent manner. These findings reinforce the importance of sex-specific analyses in pain research and highlight XPro1595 as a potential therapeutic candidate for managing oral cancer pain and PNI-associated neuropathies.

## Figures and Tables

**Figure 1 cells-14-01749-f001:**
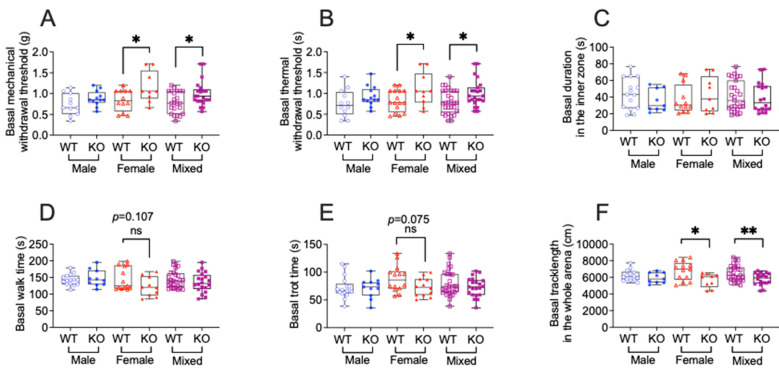
TNFR1 gene deletion affects sensory and motor function in a sex dependent manner in tumor-free mice. (**A**) Sensory behavior was assessed by a von Frey assay to determine mechanical allodynia. (**B**) Hargreaves’s test for thermal hyperalgesia. (**C**–**F**) Duration in the inner zone(s), walk time, trot time, and track length assessed by a behavior spectrometer. Mixed: male and female. WT: wild-type mice (male, *n* = 14; female, *n* = 10; mixed, *n* = 24); KO: *Tnfr1*^−/−^ mice (male, *n* = 14; female, *n* = 10; mixed, *n* = 24). Individual data points are presented for each animal. Box plots represent the interquartile range (25th–75th percentiles), with whiskers depicting the median and minimum-to-maximum values. All data were analyzed using an unpaired two-tailed *t*-test. ns = not significant (*p* > 0.05); *p* < 0.05 (*), *p* < 0.01 (**).

**Figure 2 cells-14-01749-f002:**
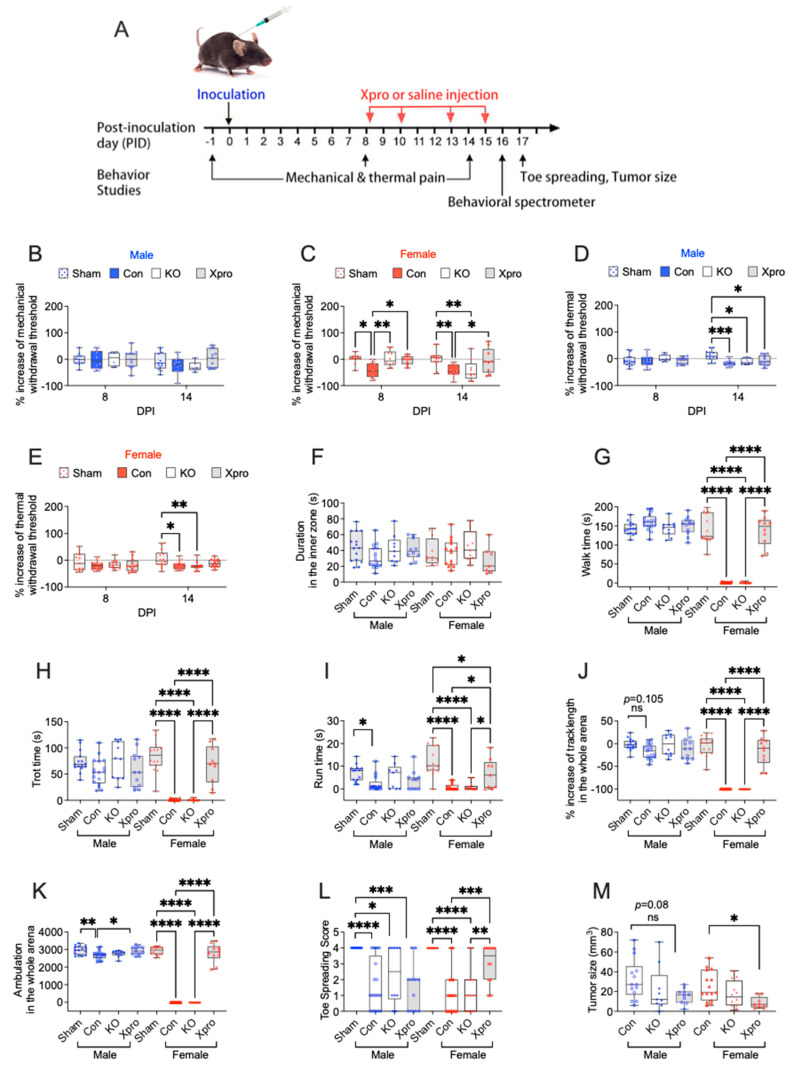
TNFR1 gene deletion or suppression ameliorate sensory and motor defects associated with PNI in a sex-dependent manner. (**A**) Schematic representation of the experimental timeline and flowchart for the sciatic nerve PNI model. The timing of TNFR1 inhibitor (XPro1595 = XPro) or vehicle (saline) administration is indicated by red arrows. On PID 17, following the toe-spreading test, mice were sacrificed and sciatic nerves, with or without tumors, were collected for tumor size measurement. (**B**,**C**) Sensory behavior was assessed by a von Frey assay to determine mechanical allodynia in male and female mice. (**D**,**E**) Hargreaves’s test for thermal hyperalgesia in male and female mice. (**F**–**K**) Duration in the inner zone (s), walk time, trot time, run time, track length, and ambulation assessed by a behavior spectrometer in male and female mice. (**L**) Toe-spreading scores. (**M**) Tumor size. Analysis was conducted to compare the effects of TNFR1 gene deletion or suppression in male and female mice. Individual data points are presented for each animal. Sham: sham-operated mice (each sex, *n* = 14); con: wild-type control tumor-bearing mice (each sex, *n* = 14); KO: Tnfr1^−/−^ tumor-bearing mice (male, *n* = 10; female, *n* = 14); XPro: XPro1595-treated wild-type tumor-bearing mice (male, *n* = 12; female, *n* = 10). Box plots represent the interquartile range (25th–75th percentiles), with whiskers depicting the median and minimum-to-maximum values. (**B**–**E**) were analyzed using two-way ANOVA followed by Tukey’s post hoc test. (**F**–**M**) were analyzed using one-way ANOVA followed by Tukey’s post hoc test. ns = not significant (*p* > 0.05); *p* < 0.05 (*), *p* < 0.01 (**), *p* < 0.001 (***), *p* < 0.0001 (****).

**Figure 3 cells-14-01749-f003:**
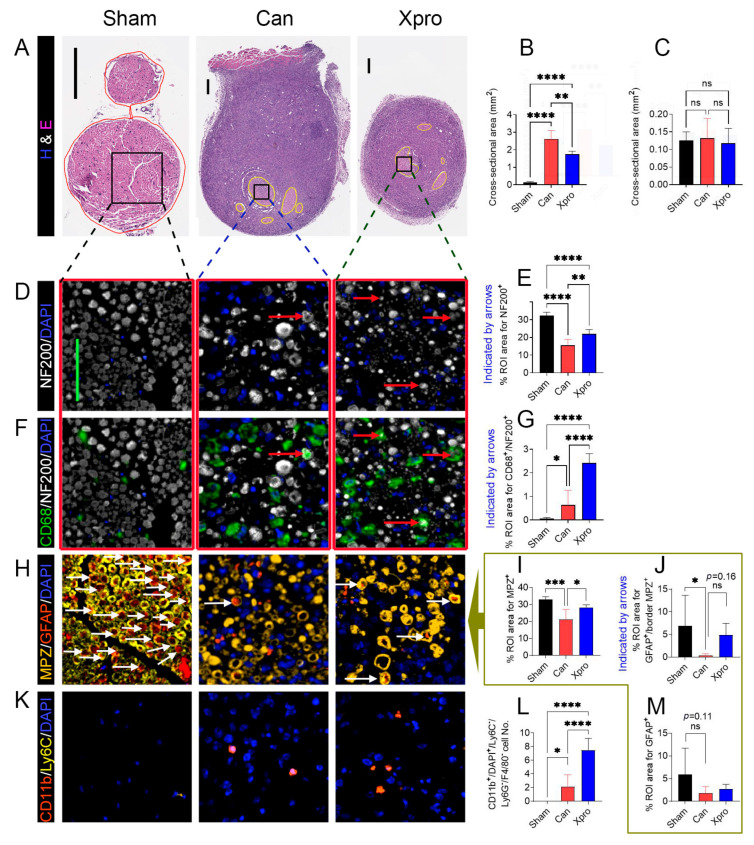
XPro1595 treatment reduced cancer-induced nerve damage in female mice. (**A**) H&E staining of the sciatic nerve cross sections. (**B**,**C**) Column bar graph analyses of cross-sectional areas of tumor in nerve zone (yellow circles) and tumor outside the nerve zone, respectively. Representative images of multiplex fluorescent staining patterns for NF200/DAPI (**D**) and quantification (**E**), CD68/NF200/DAPI (**F**) and quantification (**G**), MPZ/GFAP/DAPI (**H**) and quantification (**I**–**M**), and CD11b/Ly6C/DAPI (**K**) and quantification (**L**) in the nerve are shown. The same tissue regions, visualized with or without CD68 antibody, are indicated with red frames in (**D**,**F**). Column bar graph analyses of percent regions of interest (ROI) areas in the nerve zone relative to the tumor outside the nerve zone per mouse or cell numbers are shown in the right columns for comparison, where the column bar graphs (**I**,**J**,**M**) are analyses of images (**H**). In all figure panels, representative images are shown for sciatic nerves from the sham group (*n* = 8), Can group (*n* = 6), and XPro group (*n* = 6). Values are presented as means ± standard deviation. Statistical significance between groups was determined using one-way ANOVA followed by Tukey’s post hoc test. ns = not significant (*p* > 0.05); *p* < 0.05 (*); *p* < 0.01 (**); *p* < 0.001 (***); *p* < 0.0001 (****). Scale bars: 100 μm for (**A**); 50 μm for (**D**,**F**,**H**,**K**). Arrows shown examples of cells/axons quantified in the panel.

**Figure 4 cells-14-01749-f004:**
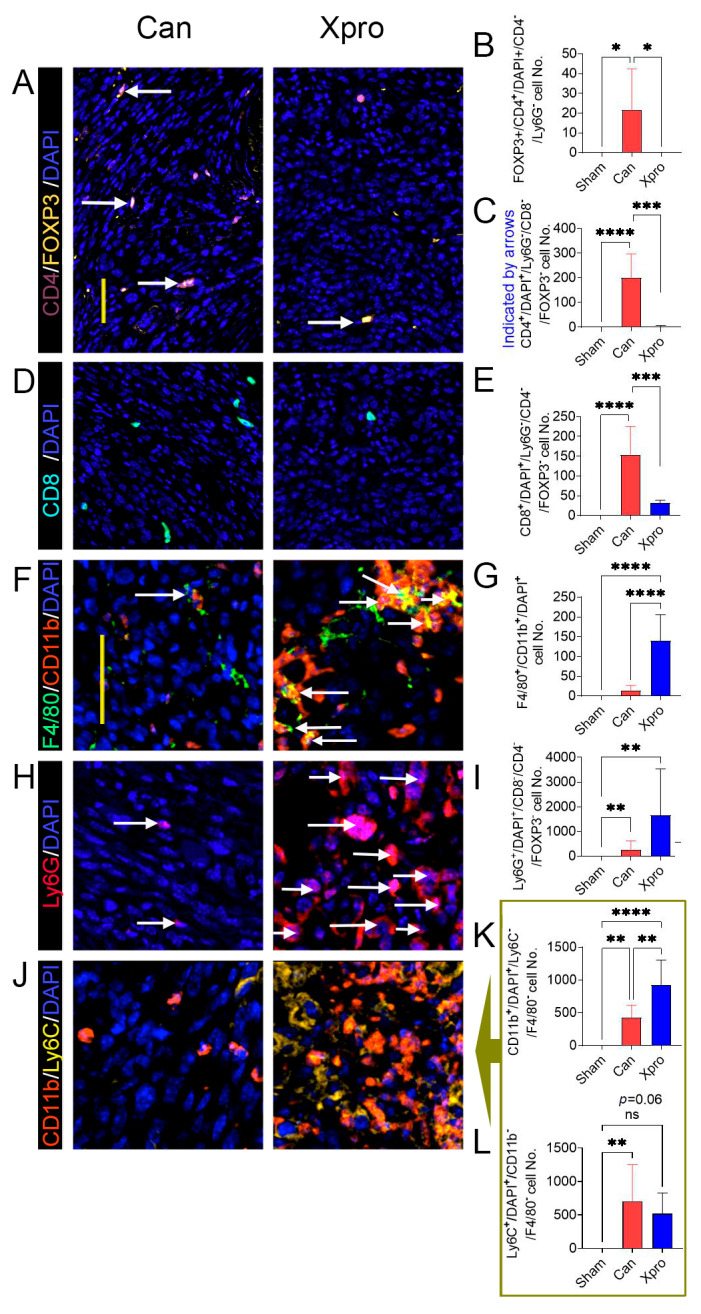
XPro1595 treatment modulates tumor-immune environment outside the nerve zone in female mice. Representative images of multiplex fluorescent staining patterns of tumor zones for CD4/FOXP3/DAPI (**A**), CD8/DAPI (**D**), F4/80/CD11b/DAPI (**F**), Ly6G/DAPI (**H**), and CD11b/Ly6C/DAPI (**J**) are shown for sciatic nerves from the sham group (*n* = 8), Can group (*n* = 6), and XPro group (*n* = 6). (**B**,**C**,**E**,**G**,**I**) are analysis of images on the left. Analysis of ROI area or cell number per mouse are shown in the right panels for comparison, where the column bar graphs (**K**,**L**) are analyses of images (**J**). Values are presented as means ± standard deviation. Data are analyzed using one-way ANOVA and the Tukey post hoc test. ns = not significant (*p* > 0.05); *p* < 0.05 (*); *p* < 0.01 (**); *p* < 0.001 (***); *p* < 0.0001 (****). Scale bar: 50 μm for (**A**,**D**) or for (**F**,**H**,**J**). White arrows: examples of cells analyzed for each panel.

**Figure 5 cells-14-01749-f005:**
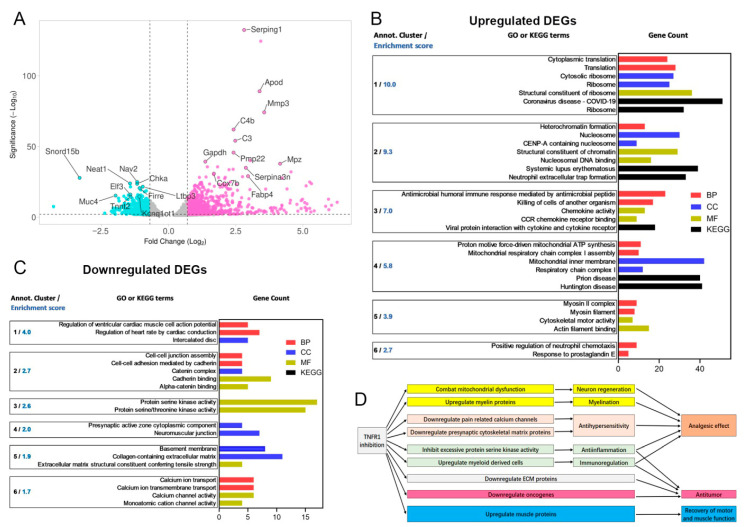
Transcriptomic analysis of the effect of XPro1595 on PNI in female mice. (**A**) A volcano plot generated with VolcaNoseR shows differentially expressed genes (DEGs) with statistical significance and fold change between the two groups. Each dot represents a gene. Statistical significance is defined as −log_10_ (adjusted *p*-value). Gray dots indicate non-significant DEGs, turquoise dots represent downregulated genes, and violet dots denote upregulated genes in XPro treated nerves with PNI vs. vehicle treated nerves with PNI. The names of the top 20 upregulated and downregulated genes are listed after ranking by Manhattan distance. (**B**,**C**) For upregulated (**B**) and downregulated (**C**) DEGs, the six most enriched clusters and representative Gene Ontology (GO) or Kyoto Encyclopedia of Genes and Genomes (KEGG) terms are displayed, along with their enrichment scores. Bars indicate GO or KEGG term enrichment and gene count. The DAVID annotation cluster analysis revealed several highly enriched GO terms (*p* < 0.05), which were grouped functionally and are also listed. These terms include GO biological process (BP, red), cellular component (CC, blue), and molecular function (MF, olive), as well as KEGG pathways (black). (**D**) Summary of the proposed mechanisms of XPro1595 based on behavioral, IF, and transcriptomic analysis.

## Data Availability

Data will be available upon reasonable request.

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
