# Peer review of "TNFR1 Suppression by XPro1595 Reduces Peripheral Neuropathies Associated with Perineural Invasion in Female Mice"

_cells, 2025, doi:10.3390/cells14221749_

Round 1

Reviewer 1 Report

Comments and Suggestions for Authors

Zhang et al. sent a original article that contais the role of TNFR1 (and its antagonism by XPro1595) in peripheral neuropathy linked to perineural invasion in female mice. This manuscript is well-written, however, must be contain some points prior to final acceptation:

  1. Check abbreviations.
  2. Figures 1C, 1D, and 1E must contain s instead of sec (international system of time = s).
  3. Can you explain better why in female the role of TNFR1 is important?. Sex hormones participate? You must explain better in discussion section the relationship between sex hormones (and other female-factors) and pain evoked by TNFR1
  4. In discussion section (paragraph 5), you mention: "XPro1595 has been shown in other models to reduce age-dependent accumulation of activated immune cells and CD4⁺ T cell popula". Is a unfinished sentence? Check it

Author Response

  1. Check abbreviations. 

We have checked abbreviations.

2. Figures 1C, 1D, and 1E must contain s instead of sec (international system of time = s).

We have changed to "s"

3.  Can you explain better why in female the role of TNFR1 is important?. Sex hormones participate? You must explain better in discussion section the relationship between sex hormones (and other female-factors) and pain evoked by TNFR1

We have added a paragraph at the end of Discussion to discuss the relationship between sex hormones and TNFR1 signaling.

4. In discussion section (paragraph 5), you mention: "XPro1595 has been shown in other models to reduce age-dependent accumulation of activated immune cells and CD4⁺ T cell popula". Is a unfinished sentence? Check it  

We have revised the sentence accordingly.  

Reviewer 2 Report

Comments and Suggestions for Authors

My comments

  1. The rationale for using both TNFR1 knockout and XPro1595 treatment groups requires clearer justification. The manuscript should explain why these two approaches were chosen and how they complement each other in testing the same hypothesis.

  2. The selection of XPro1595 dosage and administration schedule is not well justified. Information on dose optimization or reference to prior pharmacokinetic or behavioral studies should be provided to confirm that the chosen regimen is appropriate.

  3. The sex-based differences reported in behavioral outcomes are interesting, but the experimental design should clarify whether both sexes were equally represented in each group and whether estrous cycle effects were controlled.

  4. The behavioral tests, including von Frey and Hargreaves assays, appear well executed, but more details on randomization, blinding, and order of testing are needed to confirm the reliability of the results.

  5. The tumor size analysis lacks sufficient description of how tumor boundaries were defined and quantified. Providing representative images with measurement criteria would improve transparency and reproducibility.

  6. The interpretation that XPro1595 improves axonal and myelin integrity is convincing, yet additional evidence such as electron microscopy or quantification of myelin thickness would substantiate these findings.

  7. The RNA-seq analysis identifies several functional pathways, but the lack of validation by independent methods weakens the mechanistic conclusions. Verification of selected key genes at the protein level is strongly recommended.

  8. The study concludes that XPro1595 has antitumor and neuroprotective effects primarily in females, but the sample size for each subgroup seems limited. A power analysis or post hoc test of statistical robustness should be included to support this sex-specific conclusion.

Comments on the Quality of English Language

The English could be improved to more clearly express the research.

Author Response

1. The rationale for using both TNFR1 knockout and XPro1595 treatment groups requires clearer justification. The manuscript should explain why these two approaches were chosen and how they complement each other in testing the same hypothesis.

The rationale for using both TNFR1 knockout mice and XPro1595 treatment was to minimize potential limitations associated with either approach alone. Specifically, this strategy aimed to avoid off-target effects of pharmacological inhibition and control for developmental confounds inherent to gene knockout models. Consistent results from both methods would therefore strengthen the validity and reliability of the conclusions. We have now added the rationale in the Methods section, Mice.

2. The selection of XPro1595 dosage and administration schedule is not well justified. Information on dose optimization or reference to prior pharmacokinetic or behavioral studies should be provided to confirm that the chosen regimen is appropriate.

The XPro1595 dosage and administration schedule were selected based on previously published studies [1, 2] with similar experimental designs to ensure comparability across investigations.

  1. Del Rivero, T., et al., Tumor necrosis factor receptor 1 inhibition is therapeutic for neuropathic pain in males but not in females. Pain, 2019. 160(4): p. 922-931.
  2. Zhou, K.X., et al., XPro1595 ameliorates bone cancer pain in rats via inhibiting p38-mediated glial cell activation and neuroinflammation in the spinal dorsal horn. Brain Res Bull, 2019. 149: p. 137-147.

The references are added to Results section, 3.2.

3. The sex-based differences reported in behavioral outcomes are interesting, but the experimental design should clarify whether both sexes were equally represented in each group and whether estrous cycle effects were controlled.

The sex-based differences observed in behavioral outcomes were derived from data including at least ten mice per sex and per test. We believe that both sexes were adequately represented within each group. However, estrous cycle stages were not monitored, as the meta-analysis on how estrous cycle affects the pain behavior is still debatable and it is context-dependent (https://onlinelibrary.wiley.com/doi/full/10.1002/ejp.714).

4. The behavioral tests, including von Frey and Hargreaves assays, appear well executed, but more details on randomization, blinding, and order of testing are needed to confirm the reliability of the results.

Mice are randomly assigned to experimental groups; testing was conducted with blinding (Methods). Order of testing are shown in Methods and Results, 3.1 and figure legends.

5. The tumor size analysis lacks sufficient description of how tumor boundaries were defined and quantified. Providing representative images with measurement criteria would improve transparency and reproducibility.

Tumor area was measured using the nerve-cross section area, confirmed the presence of tumor using H&E, a clinically relevant approach. Representative images were shown in Fig. 3A. We have added a sentence in the Method section, Multiplex immunohistochemistry and immunofluorescence (mIHC/IF) analysis.

6. The interpretation that XPro1595 improves axonal and myelin integrity is convincing, yet additional evidence such as electron microscopy or quantification of myelin thickness would substantiate these findings.

We have both RNAseq data and IF imaging to show that axon and myelin integrity is compromised which we believe is sufficient.

7. The RNA-seq analysis identifies several functional pathways, but the lack of validation by independent methods weakens the mechanistic conclusions. Verification of selected key genes at the protein level is strongly recommended.

We have used the Multiplex immunohistochemistry and immunofluorescence (mIHC/IF) analysis to validate the RNA-seq findings. For example, we showed that genes encoding myelin associated proteins (e.g. Mpz) are downregulated, and in the IF analysis we also showed a reduction in myelin protein MPZ.

8. The study concludes that XPro1595 has antitumor and neuroprotective effects primarily in females, but the sample size for each subgroup seems limited. A power analysis or post hoc test of statistical robustness should be included to support this sex-specific conclusion.

The sample size was determined based on our previous empirical data and related studies. Power analysis indicated that the behavioral spectrometer assay required the largest group size (n = 10); therefore, each group included at least 10 mice. Results of post hoc statistical analyses are presented in the corresponding figures.

We have added a sentence in the Method section, The mouse PNI model.

Round 2

Reviewer 2 Report

Comments and Suggestions for Authors

No comments

Comments on the Quality of English Language

The English could be improved to more clearly express the research.